# Soluble Epoxide Hydrolase as a Therapeutic Target for Neuropsychiatric Disorders

**DOI:** 10.3390/ijms23094951

**Published:** 2022-04-29

**Authors:** Jiajing Shan, Kenji Hashimoto

**Affiliations:** Division of Clinical Neuroscience, Chiba University Center for Forensic Mental Health, Chiba 260-8670, Japan; shanjiajing0525@gmail.com

**Keywords:** autism spectrum disorder, depression, inflammation, Parkinson’s disease, schizophrenia, soluble epoxide hydrolase, stroke

## Abstract

It has been found that soluble epoxide hydrolase (sEH; encoded by the *EPHX2* gene) in the metabolism of polyunsaturated fatty acids (PUFAs) plays a key role in inflammation, which, in turn, plays a part in the pathogenesis of neuropsychiatric disorders. Meanwhile, epoxy fatty acids such as epoxyeicosatrienoic acids (EETs), epoxyeicosatetraenoic acids (EEQs), and epoxyeicosapentaenoic acids (EDPs) have been found to exert neuroprotective effects in animal models of neuropsychiatric disorders through potent anti-inflammatory actions. Soluble expoxide hydrolase, an enzyme present in all living organisms, metabolizes epoxy fatty acids into the corresponding dihydroxy fatty acids, which are less active than the precursors. In this regard, preclinical findings using sEH inhibitors or *Ephx2* knock-out (KO) mice have indicated that the inhibition or deficiency of sEH can have beneficial effects in several models of neuropsychiatric disorders. Thus, this review discusses the current findings of the role of sEH in neuropsychiatric disorders, including depression, autism spectrum disorder (ASD), schizophrenia, Parkinson’s disease (PD), and stroke, as well as the potential mechanisms underlying the therapeutic effects of sEH inhibitors.

## 1. Introduction

Neuropsychiatric disorders, including depression, autism spectrum disorder (ASD), schizophrenia, and Parkinson’s disease (PD), are common brain diseases characterized by cognitive deficits, psychiatric symptoms, and somatoform symptoms [1]. Meanwhile, psychiatric disorders such as depression, ASD, and schizophrenia are cumulatively common and show a remarkable increase of prevalence in young people [2,3]. Moreover, according to the 2019 Global Burden of Diseases, Injuries, and Risk Factors Study (GBD), psychiatric disorders are among the leading causes of the global health-related burden [4]. This burden was generally high across the entire lifespan for both genders and across many locations [5].

In 2020, as COVID-19 spread throughout the world, the increasing SARS-CoV-2 infection rates and decreasing human mobility contributed to a significant increase in the prevalence of neuropsychiatric disorders [6]. Neurological disorders, including PD, Alzheimer’s disease, and stroke, have also become a major public health challenge, due to the decreased quality of life and increased burden for millions of patients and their caregivers [7]. Although several approved drugs have been used to eliminate some symptoms in patients with neuropsychiatric disorders, the development of new therapeutic drugs for such disorders must occur to meet the medical needs of individuals [8,9,10,11,12].

In general, fatty acids are the major products of lipid metabolism [13]. Additionally, polyunsaturated fatty acids (PUFAs) are essential dietary fats that include more than one double bond and are classified based on the number of carbon atoms located between the first double bond of the carbon chain and the terminal methyl end [14,15]. In particular, PUFAs in the n-3 and n-6 (omega-3 and omega-6) families play a prominent role in biological function as components of cell membranes, membrane fluidity regulation, membrane-associated proteins, and neurotransmission [16]. Increasing evidence has shown that inadequate diets or metabolic deficiencies can cause low levels of n-3 PUFAs, which are related to the etiologies of various neuropsychiatric disorders such as PD, schizophrenia, depression, and attention deficit hyperactivity disorder [14,17,18,19].

The biologically important long-chain PUFAs include docosahexaenoic acid (DHA or 22:6 n-3) (see Figure 1), which represents approximately 40% of central PUFAs. Such acids contribute to neural cell signaling, membrane fatty acid chain fluidity, ion permeability, and protein function [20,21]. At the cerebral level of the n-6 family, roughly 50% of PUFAs are represented by arachidonic acid (ARA, 20:4 n-6) (see Figure 1), which plays a role in signaling, memory, and learning modulation [22,23]. Another important lipid is eicosapentaenoic acid (EPA, 20:5 n-3) (see Figure 1), whose concentration is significantly lower in the central nervous system. It also stands out for its anti-inflammatory properties, which can have neuroprotective impacts on the brain [24,25].

Specifically, PUFAs are metabolized into bioactive derivatives by main enzymes such as cyclooxygenases (COXs), lipoxygenases (LOXs), and cytochrome P450s (CYPs) (see Figure 1) [26,27,28], whereas EPA is converted into hydroxyeicosapentaenoic acids (HEPEs) and prostaglandin E3 (PGE3) through the LOX and COX pathways, respectively. Moreover, neuroprotectin D1 (NPD1) and electrophile oxo-derivatives (EFOXs) are synthesized from DHA through the LOX and COX pathways, respectively. These mediators such as HEPEs, PGE3, NPD1, and EFOXs act as anti-apoptotic protein activators and suppress inflammatory gene expression (see Figure 1) [14]. EPA and DHA are also converted into epoxyeicosatetraenoic acids (EEQs) and epoxydocosapentaenoic acids (EDPs) through the CYP pathway, respectively. These epoxide fatty acids are then metabolized into their corresponding diols (DHETEs and DHDPAs) by soluble epoxide hydrolase (sEH: coded by the *EPHX2* gene). For example, 19,20-epoxydocosapentaenoic acid (19,20-EDP) synthesized from DHA through the CYP pathway is metabolized into 19,20-dihydroxydocosapentaenoic acid (19,20-DHDPA) by sEH [29,30]. ARA is also metabolized by the COX and LOX pathways to create a class of compounds known as leukotrienes and prostaglandins, which are important signaling molecules that control pro-inflammatory actions (see Figure 1 and Figure 2) [26,27,31]. Furthermore, the CYP pathway generates epoxyeicosatrienoic acids (EETs) from ARA, and EETs are metabolized into dihydroxyeicosatrienoic acids (DHETs) by sEH (see Figure 1 and Figure 2) [26,27,32].

Since DHETs dramatically reduce biologic activity, sEH inhibitors have been extensively used to prolong the anti-inflammatory function of EETs in the ARA cascade [33,34]. Such inhibitors have also been shown to decrease sEH activity, with little to no toxicity in animal models [35,36]. Meanwhile, inflammatory cytokines and chemokines are found in various neurobiological pathways which are related to neuropsychiatric disorders [37]. Overall, it is likely that sEH plays a role in the pathogenesis of psychiatric and neurological disorders. Thus, this review discusses the role of sEH in neuropsychiatric disorders such as depression, autism spectrum disorder (ASD), schizophrenia, Parkinson’s disease (PD), and stroke as well as the potential mechanisms underlying the therapeutic effects of sEH inhibitors.

## 2. Depression

According to the 2021 Global Health Data Exchange, depression is one of the most common psychiatric disorders in the world, with an estimated 3.8% of the population affected, including 5.0% of adults and 5.7% of adults 60 years and older [38]. Current antidepressants such as selective serotonin reuptake inhibitors or serotonin–norepinephrine reuptake inhibitors can take several weeks before they are effective. Since approximately one-third of patients with depression do not respond to current antidepressants, new antidepressants must be developed for such treatment-resistant patients [27,39,40]. Moreover, numerous studies have suggested a strong association between inflammatory processes and the pathophysiology of depression [41,42,43,44,45,46,47].

The cytochrome P450 epoxygenase CYP2J2 converts ARA into four regioisomeric EETs (see Figure 2), while systemic overexpression of human CYP2J2 reduces the increased plasma levels of inflammatory cytokines and decreased levels of the anti-inflammatory mediator interleukin-10 (IL-10) after injection of tumor necrosis factor-α (TNF-α) [48]. In addition, the increase of inflammatory protein in TNF-α treated human bronchi is suppressed by 14,15-EET [49]. These findings suggest that the decrease of EETs metabolized by sEH can aggravate inflammation in the brain. Hence, regarding inflammation in depression, sEH is likely to play a crucial role [27,39].

In a related study, the expression of sEH protein in the brain was higher in a sample of susceptible mice after chronic social defeat stress (CSDS) compared to the control mice [50]. Other findings from the study were as follows. First, the expression of sEH protein was higher in the postmortem brain samples of patients with depression compared to those of the controls [50]. Second, pretreatment with the sEH inhibitor TPPU [1-(1-propionylpiperidin-4-yl)-3-(4-(trifluoromethoxy) phenyl) urea] prevented the onset of depression-like behaviors after CSDS. Third, the sample of *Ephx2* KO mice did not show depression-like behaviors after CSDS, suggesting stress resilience [50]. Interestingly, fecal microbiota transplantation from CSDS-susceptible mice with depression-like phenotype produced such a phenotype in antibiotic-treated *Ephx2* KO mice, indicating that the administration of “depression-related microbes” can contribute to the conversion of resilient *Ephx2* KO mice into KO mice with depression-like phenotype [51]. Altogether, these results suggest that sEH plays a key role in the pathophysiology of depression and that sEH inhibitors can be potential therapeutic or prophylactic drugs for depression [27,39,50].

The downregulation of hepatic sEH in mice caused a reduction in sucrose preference and coat deterioration compared with the control group [52]. Moreover, patients with depression showed higher levels of sEH protein in the parietal cortex and liver compared to those in the control group [53]. Thus, it is likely that the brain–liver axis plays a role in depression [52,53,54].

Immunoreactivity of sEH was also detected in astrocytes throughout the brain [55], whereas sEH activity in the astrocytes of the medial prefrontal cortex (PFC) of CSDS-susceptible mice was negatively correlated with depression-like behaviors [56]. Moreover, a recent study showed that sEH in the central nucleus of the amygdala regulates anxiety-related behaviors [57], while TPPU produced antidepressant-like effects in the lipopolysaccharide (LPS)-induced inflammation model of depression and in the CSDS model [50]. However, the antidepressant-like effects of TPPU were blocked by the tropomyosin receptor kinase B (TrkB) antagonist, indicating that brain-derived neurotrophic factor (BDNF)-TrkB signaling plays a certain role in the antidepressant-like effects of TPPU [58,59].

In other research, pretreatment with TPPU attenuated the increase of pro-inflammatory cytokine IL-1β and rescued neuronal and dendritic spine loss in the hippocampus by increasing the expression of the N-methyl-D-aspartate receptor, the extracellular-signal-regulated kinase (ERK)1/2, and the CREB (cAMP response element binding protein) [60]. In the LiCl-pilocarpine-induced post-status epilepticus rat model, TPPU attenuated spontaneous recurrent seizures and epilepsy-associated depression-like behaviors through anti-inflammatory effects [61], while co-treatment with TPPU, EPA, and DHA was more effective in preventing IL-1β, IL-6, and TNF-α-induced decreased neurogenesis and increased apoptosis [62]. Furthermore, the serum levels of sEH-derived fatty acid diols increased in depressed patients with type 2 diabetes mellitus, while depressive symptom severity was associated with the oxylipin profile [63], suggesting higher activity of sEH in these patients. In sum, increased activity of sEH most likely plays a role in the pathogenesis of depression and suggests that sEH inhibitors are potential antidepressants [27,39,64,65,66,67].

## 3. ASD and Schizophrenia

Accumulating evidence has suggested that maternal immune activation (MIA) such as maternal infection can increase the risk of neuropsychiatric disorders (e.g., ASD and schizophrenia) in offspring [68,69,70,71,72,73,74,75]. A meta-analysis also showed a strong relationship between maternal infection during pregnancy and the increased risk of ASD in offspring [76]. It was also pointed out that the COVID-19 pandemic may increase the risk of ASD and schizophrenia in offspring after maternal infection of SARS-CoV-2 [77,78,79].

Meanwhile, MIA using poly(I:C) has been widely used as animal models of ASD and schizophrenia [80,81]. For example, using rodents, there were higher levels of sEH and decreased levels of epoxy fatty acids (i.e., 10,11-EDP, 5,6-EET, 8,9-EET, 11,12-EET) in the PFC of juvenile offspring after MIA, indicating increased activity of sEH in the PFC of juvenile offspring after MIA [82]. The expression of *EPHX2* mRNA in induced pluripotent stem cell-derived neurospheres was higher among schizophrenia patients than the controls [82]. There was also a higher expression of *EPHX2* mRNA in the postmortem brain samples of ASD patients than that of the controls [82]. Additionally, the levels of sEH in the parietal cortex from schizophrenia patients were higher than those of the controls [50,53]. Collectively, neuroinflammation by the increased expression of sEH most likely plays a role in the pathogenesis of ASD and schizophrenia.

In related research, repeated treatment with TPPU in juvenile offspring from prenatal days P28 to P56 prevented cognitive deficits and loss of parvalbumin (PV)-immunoreactivity in the medial PFC of adult offspring, especially after MIA [82]. Additionally, treatment with TPPU in pregnant mothers from E5 to P21 prevented cognitive deficits, social interaction deficits, and PV-immunoreactivity in the medial PFC of juvenile offspring after MIA. Altogether, increased activity of sEH in the brain can contribute to ASD (or schizophrenia)-like behavioral abnormalities in offspring after MIA.

Epidemiological studies have suggested that exposure to herbicides (i.e., glyphosate) during pregnancy might increase the risk of ASD in offspring. For instance, a population-based case study in California (USA) reported that the risk of ASD was associated with the use of glyphosate (odds ratio = 1.16) [83]. It was also found that maternal glyphosate exposure during pregnancy and lactation caused ASD-like behavioral abnormalities, an increase of expression of sEH in the PFC, hippocampus, and striatum of juvenile offspring, and a decrease of PV-immunoreactivity in the prelimbic of the medial PFC of juvenile mice [84,85]. In addition, the levels of 8,9-EET in the blood and brain regions (i.e., PFC, hippocampus, and striatum) of juvenile offspring after maternal glyphosate exposure were lower than those of the control groups, indicating increased expression of sEH in the brain regions [84]. Moreover, oral administration of TPPU to pregnant mothers from E5 to P21 prevented ASD-like behaviors such as social interaction deficits and increased grooming time in the juvenile offspring after maternal glyphosate exposure. Collectively, increased sEH in the brain seems to play a role in the pathogenesis of ASD after maternal glyphosate exposure [84,85,86].

Finally, the potent sEH inhibitor AS2586114 improved schizophrenia-like behavioral abnormalities (e.g., hyperlocomotion and pre-pulse inhibition deficits) in a sample of phencyclidine (PCP)-treated mice, suggesting that sEH inhibitor might have antipsychotic-like activity [87]. Furthermore, TPPU in drinking water during the juvenile and adolescent stages of offspring can also prevent the onset of cognitive deficits and a reduction of PV-immunoreactivity in the medial PFC of adult offspring after MIA [82]. Repeated treatment with TPPU from pregnancy to weaning can also prevent the onset of cognitive deficits in juvenile offspring after MIA or maternal glyphosate exposure [82,84]. Overall, these findings indicate that sEH plays a key role in the development of ASD and schizophrenia in offspring after MIA and that sEH inhibitors can have prophylactic or therapeutic impacts on neuropsychiatric disorders [10,88,89].

## 4. Parkinson’s Disease

Parkinson’s disease (PD) is a neurodegenerative disease characterized by the deposition of the aggregates of α-synuclein (termed “Lewy bodies”) and the loss of dopaminergic neurons in the substantia nigra (SN), which results in motor dysfunction and non-motor dysfunction [90,91,92]. PD is the second most prevalent neurodegenerative disorder, only after Alzheimer’s disease. It is also predicted that the number of patients with PD will double over the next 20 years [93]. Meanwhile, L-DOPA (the precursor of dopamine) or dopamine (DA) receptor agonists have been used in the treatment of PD [94]. Although these treatments seem to alleviate symptoms, there are no disease-modifying or neuroprotective drugs for PD [95,96,97].

Multiple evidence has supported neuroinflammation-related oxidative stress in the pathogenesis of PD [66,98,99,100]. Related research has shown that activated astrocytes and microglia caused by brain immune response are involved in the development of neuroinflammatory features, leading to the exacerbation of DA neurons in the substantia nigra pars compacta (SNc) [101]. In addition, EETs, regulators of inflammation processes, can produce the neurotrophic role of astrocytes, increase the release of BDNF, reduce glutamatergic toxicity through the astrocytic metabotropic glutamate receptor mGluR5, prevent mitochondrial dysfunction and apoptosis, and protect synaptic function in the brain [100,102,103,104].

MPTP (1-methyl-4-phenyl-1,2,3,6-tetrahydropyridine)-induced neurotoxicity in the striatum and SNc of a sample of rodents has been a well-established preclinical in vivo model to study the pathogenesis of PD [105,106]. Higher expression of the sEH protein in the striatum has also been found in MPTP-treated mice or postmortem brain samples from dementia patients with Lewy bodies [107]. Moreover, the expression of the *EPHX2* mRNA in human PARK2 (Parkinson’s disease protein 2) iPSC-derived neurons were significantly higher compared to the control groups [107], while the levels of 8,9-EET in the striatum of MPTP-treated mice were lower than those of the control mice, suggesting higher activity of sEH in the striatum of the MPTP-treated mice [107]. Interestingly, sEH expression was positively correlated to the phosphorylation of α-synuclein in the striatum of the MPTP-treated mice [107]. Since PD may not display symptoms until approximately 80% of the striatal DA has been lost, a biomarker for earlier diagnosis is important for finding treatment options [108]. In this regard, peripheral detection of elevated levels of sEH in the gut, which is possible prior to the losses of dopaminergic neurons, may potentially provide an early biomarker of PD [95,108].

MPTP-induced neurotoxicity in the striatum and SN, including the loss of the DA transporter (DAT), loss of tyrosine hydrolase (TH)-positive cells, and increased endoplasmic reticulum (ER) stress, was attenuated by repeated oral administration of TPPU [107]. Additionally, both AUDA [12-(((tricyclo(3.3.1.13,7)dec-1-ylamino)carbonyl)amino)-dodecanoic acid], another sEH inhibitor, and sEH deficiency significantly protected against MPTP-induced toxicity [109,110]. A recent study found that the natural compound kurarinone, an uncompetitive inhibitor of sEH, can alleviate the MPTP-induced behavioral deficits, dopaminergic neurotoxicity, and neuroinflammation via suppressing the activated microglia, including the nuclear factor kappa B (NF-κB) signaling pathway [111]. Furthermore, the inhibition of sEH can suppress the aggregation of α-synuclein, microglia activation, neuroinflammation, and apoptosis, resulting in the neuroprotection activity in the PD model [112,113]. Altogether, increased sEH and the resulting increase in phosphorylation of α-synuclein may play a role in the pathogenesis of PD and that sEH inhibitors can be potential drugs for PD.

## 5. Stroke

Stroke, constituting a loss of blood flow to the brain, is the leading cause of disability and death worldwide. In fact, the global prevalence of stroke in 2019 alone was 101.5 million people [114]. Acute episodes, including ischemic strokes, hemorrhagic strokes, traumatic brain injury, and seizures, can cause neuroinflammation in the brain, resulting in the loss of neurons and activation of resident immune cells [92]. Subsequent activation of immune responses can increase pro-inflammatory cytokines [115] and compromise the brain-blood barrier (BBB), further deteriorating neurodegeneration and exacerbating the injury caused by stroke [116].

Despite extensive efforts to discover better therapies for stroke, treatment options are still limited. The primary current treatment for ischemic stroke is thrombolysis with the tissue plasminogen activator (tPA), which was approved by the Food and Drug Administration. However, it must be administered within a relatively short time window before neuronal loss occurs [117]. While the tPA can be effective for patients with ischemic stroke, it can also aggravate hemorrhagic strokes, which include similar clinical symptoms with ischemia [118]. Thus, developing a therapy that focuses on a single agent that targets multiple mechanisms of ischemic brain injury may prove more effective [119].

As for sEH, it catalyzes the metabolism of EETs, which exhibit potentially beneficial actions in stroke via vasodilation, neuroprotection, promotion of angiogenesis, suppression of platelet aggregation, oxidative stress, and post-ischemic inflammation [120]. In related research, hypertension- and diabetes-associated ischemic stroke risk increased through the *EPHX2* gene variants in the Turkish population [121]. Moreover, serum oxylipin changes consistent with higher sEH activity played a key role in vascular cognitive impairment, which is associated with the injury of periventricular subcortical white matter [122]. Collectively, sEH inhibition is a potential intervention for stroke patients based on the beneficial properties of EETs.

The inhibition of sEH is also effective for reducing infarct volume, improving memory deficits, and alleviating cognitive impairment and microvasculature augmentation by suppressing neuroinflammation and increasing reparative cytokines and growth factors such as BDNF and doublecortin [123,124,125]. Regarding BDNF, it activates the receptor TrkB to promote the growth and differentiation of nerve cells and to have a neuroprotective effect on the neurons. Thus, a blockade of the BDNF-TrkB signaling pathway by the TrkB inhibitor ANA-12 can abolish the protective effect of sEH gene deletion in the middle cerebral arterial occlusion (MCAO) models of stroke [126], suggesting the role of BDNF-TrkB signaling in the actions of sEH inhibitors.

Finally, administration of TPPU or AUDA significantly promoted M2 polarization of microglial cells, which indicates a shift from pro-inflammatory polarized microglia to microglia that primarily release anti-inflammatory cytokines. This can result in the differentiation of oligodendrocytes, protection against white matter integrity, and remyelination against chronic hypoperfusion [127,128,129,130,131]. Interestingly, treatment with t-AUCB (a sEH inhibitor) after ischemic stroke onset has been shown to exert brain protection in a sample of non-diabetic mice, but not in type 2 diabetes mellitus (DM2) mice, while DM2-induced hyperglycemia can abolish t-AUCB-mediated neuroprotection against stroke [132]. SMTP-7 targeting sEH has also been shown to be effective in treating severe embolic stroke in a sample of monkeys under conditions in which tPA treatment causes hemorrhagic infarct-associated premature death [133,134]. Furthermore, related research has shown that treatment with the sEH inhibitor can decrease the activity of matrix metalloproteases (MMP)-2 and MMP-9, increase the expression of tight junction proteins, reduce activation of NF-κB, and suppress the apoptosis to protect the BBB integrity from ischemia [135,136]. Altogether, these studies suggest that sEH inhibition can have multi-target protective effects and alleviate cognitive impairment after a stroke.

## 6. Conclusions and Future Perspectives

As discussed in this review, sEH inhibitors can exert a neuroprotective effect through potent anti-inflammatory actions, including BDNF-TrkB activation for inflammation-related endoplasmic reticulum stress and mitochondrial dysfunction (Figure 3). For example, EETs suppress the MMP-2 and MMP-9, resulting in the prevention of mitochondrial dysfunction. Furthermore, EETs induce the expression of BDNF which binds to its receptor TrkB, resulting in the MEK–ERK–CREB signaling pathway [45,137]. BDNF–TrkB–MEK–ERK–CREB pathway could contribute to neurite outgrowth, neurogenesis, and synaptic plasticity. Furthermore, EETs could reduce oxidative stress and ER stress through suppression of NF-kB and activation of Bcl-2, respectively. In addition, EETs can protect against apoptosis through suppression of the BAX (Bxcl-2-associated X protein) and caspase-3 (see Figure 3). Thus, it is likely that sEH inhibitors can potentially serve as prophylactic or therapeutic drugs for neuropsychiatric disorders such as depression, ASD, schizophrenia, PD, and stroke.

Currently, clinical studies using two sEH inhibitors (i.e., GSK2256294 and EC5026) (see Figure 4) in humans are underway [138]. For instance, a recent randomized, double-blind, placebo-controlled study found that treatment with GSK2256294 for seven days in a sample of subjects with obesity and prediabetes (n = 16) effectively inhibited sEH activity in plasma, muscle, and adipose tissue. Although it reduced F2-isoprostanes (a marker of oxidative stress), it did not improve insulin sensitivity or blood pressure [139]. Moreover, a double-blind, randomized placebo-controlled trial indicated that treatment with GSK2256294 for 10 days in a sample of patients (n = 10) with aneurysmal subarachnoid hemorrhage resulted in a considerable increase in the serum EET/DHET ratio at days 7 and 10 but not in the cerebrospinal fluid (CSF) [140]. Conversely, there was decreased CSF inflammatory cytokines following GSK2256294 treatment, but it did not achieve statistical significance [140]. These clinical studies showed that treatment with GSK2256294 can cause an increase in the EET/DHET ratio of blood and tissue through potent sEH inhibition in humans. Furthermore, although a clinical trial of EC5026 in humans is underway by EicOsis Human Health Inc. (Davis, CA, USA) [138], future randomized, double-blind, placebo-control studies using a large sample size are necessary for determining the efficacy of sEH inhibitors in patients with neuropsychiatric disorders.

## Figures and Tables

**Figure 1 ijms-23-04951-f001:**
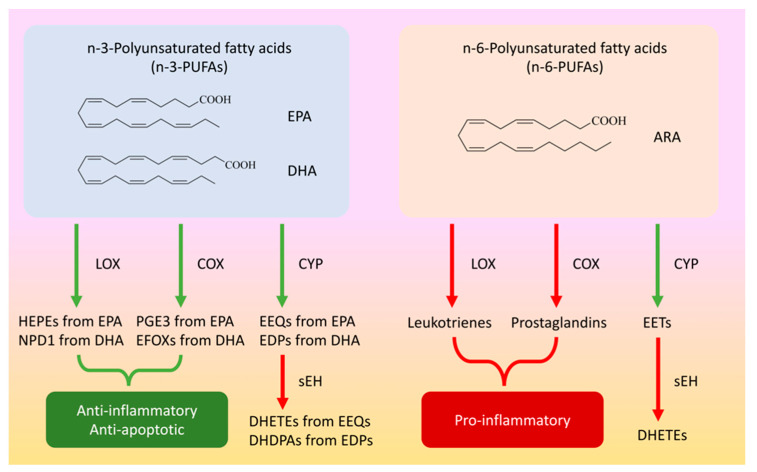
Metabolism of eicosapentaenoic acid (EPA), docosohexaenoic acid (DHA), and arachidonic acid (ARA). Eicosapentaenoic acid (EPA) is converted into hydroxyeicosapentaenoic acids (HEPEs) and prostaglandin E3 (PGE3) through the lipoxygenase (LOX) and cyclooxygenase (COX) pathways, respectively. Docosohexaenoic acid (DHA) is also converted into neuroprotectin D1 (NPD1) and electrophile oxo-derivatives (EFOXs) through the LOX and COX pathways, respectively. The compounds HEPEs, NPD1, PGE3, and EFOXs act as anti-inflammatory and anti-apoptotic mediators, while EPA and DHA are converted into epoxyeicosatetraenoic acids (EEQs) and epoxydocosapentaenoic acids (EDPs) through the cytochrome P450 (CYP) pathway, respectively. Moreover, these epoxide fatty acids are metabolized into their corresponding diols (DHETEs and DHDPAs) by soluble epoxide hydrolase (sEH), while arachidonic acid (ARA) is converted into leukotrienes and prostaglandins by the LOX and COX pathways, respectively. Finally, ARA is converted into epoxyeicosatrienoic acids (EETs) by the CYP pathway, and these EETs are metabolized into their corresponding diols (DHETs) by sEH. In this case, epoxy fatty acids (EEQs, EDPs, and EETs) have anti-inflammatory effects.

**Figure 2 ijms-23-04951-f002:**
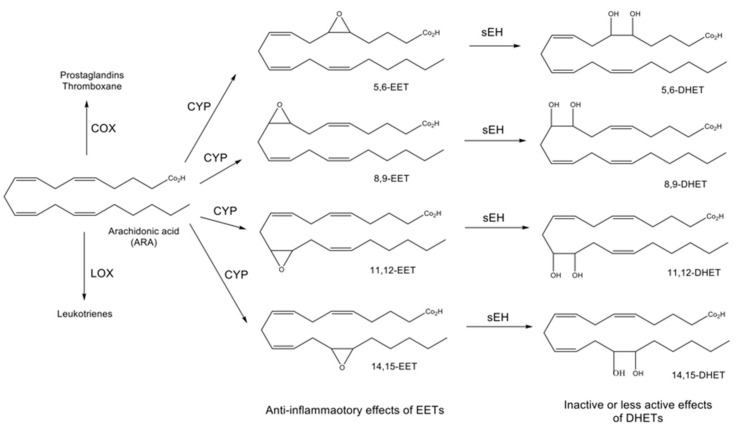
Roles of CYP and sEH in the arachidonic acid (ARA) cascade. In addition to the COX and LOX pathways, ARA is metabolized into four epoxyeicosatrienoic acids (EETs) through the cytochrome P450 (CYP) pathway. EETs are then metabolized into the corresponding dihydroxyeicosatetraenoic acids (DHETs) by soluble epoxide hydrolase (sEH). EETs have anti-inflammatory effects, while DHETs are inactive or have fewer active effects.

**Figure 3 ijms-23-04951-f003:**
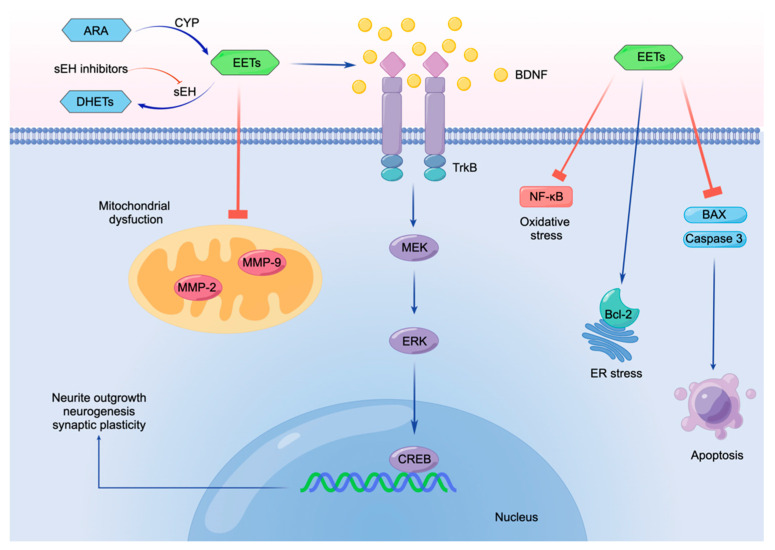
Molecular mechanisms underlying the neuroprotective effect of EETs. EETs suppress the MMP-2 and MMP-9 to prevent mitochondrial dysfunction. EETs induce the expression of BDNF, then stimulate BDNF-TrkB signaling. Subsequently, BDNF-TrkB signaling stimulates the MEK–ERK–CREB signaling pathway, promoting neurite outgrowth, neurogenesis, and synaptic plasticity. Furthermore, EETs reduce oxidative stress and suppress ER stress through suppression of NF-kB and activation of Bcl-2, respectively. Moreover, EETs protect against apoptosis through suppressing the BAX and Caspase-3; MMP-2: matrix metalloproteinase-2; MMP-9: matrix metalloproteinase-9; BDNF: brain derived neurotrophic factor; TrkB: tropomyosin receptor kinase B; MEK: mitogen-activated protein kinase; ERK: extracellular signal-regulated kinase; CREB: cAMP response element binding protein; ER: endoplasmic reticulum; NF-κB: nuclear factor kappa-light-chain-enhancer of activated B cells; Bcl-2: B-cell lymphoma 2; BAX: Bcl-2-associated X protein. Made by Figdraw (www.figdraw.com, accessed on 24 March 2022).

**Figure 4 ijms-23-04951-f004:**
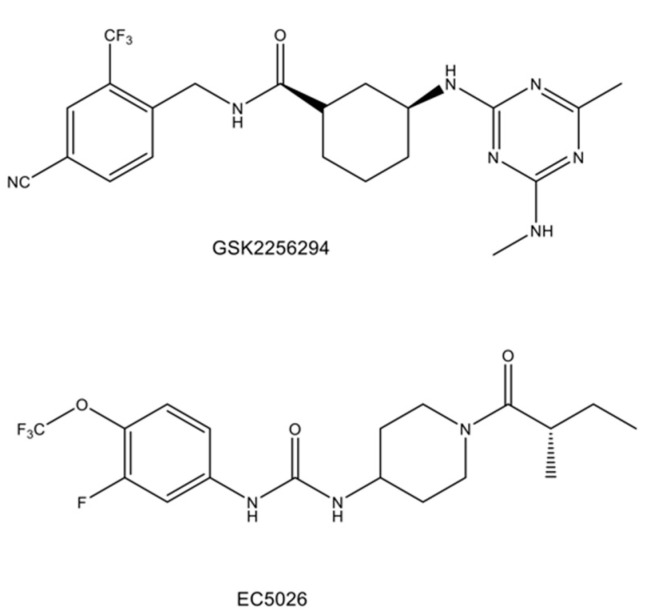
Chemical structure of GSK2256294 and EC5026 for human use.

## Data Availability

Not applicable.

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
