# Peer review of "Soluble Epoxide Hydrolase as a Therapeutic Target for Neuropsychiatric Disorders"

_ijms, 2022, doi:10.3390/ijms23094951_

Round 1

Reviewer 1 Report

The paper is well written with information clearly presented. It is evident that the manuscript has been carefully proofread and revised before submission. The manuscript provides a clear and comprehensive review on soluble epoxide hydrolase as a therapeutic target for neuropsychiatric disorders. I recommend publication subject to some minor issues being addressed. Figures 1-3 captions are very long (between 140-180 words in length). Most of the text in these captions would be better incorporated in the main body of the text. In my opinion Figure 4 does not really add anything to the paper and should be deleted.

Author Response

We revised the manuscript according to the comments by the reviewer #1. We deleted the old Figure 4.

Reviewer 2 Report

The account by Shan and Hashimoto discusses the metabolism of polyunsaturated fatty acids, focusing on epoxy fatty acids (EETs) known to have anti-inflammatory effects. Since the diols derived from their metabolism (DHETs) are less active, the enzyme responsible for this hydrolysis, the soluble epoxide hydrolase (sEH), plays a pivotal role in inflammation. More interestingly, the authors point out on the connection between sEH activity and several neuropsychiatric disorders, discussing the most relevant pre-clinical studies reported in the literature for depression, autism spectrum disorder, schizophrenia, Parkinson’s disease, and stroke. In the conclusion and future perspectives, the authors describe the clinical studies in humans that are currently ongoing regarding the administration of sEH inhibitors to increase the EET/DHET ratio.

The topic is of great interest and the manuscript is clear, well written and well organized and the publication on International Journal of Molecular Sciences is recommended after minor aspects will be improved.

  • At the end of the Introduction section a sentence claiming the aim of the review and the disorders that will be addressed should be added. Indeed, such a sentence is already present but in the caption of Figure 2 (lines 107-110): “Thus, this review discusses the role of sEH in neuropsychiatric disorders, such as depression, autism spectrum disorder (ASD), schizophrenia, Parkinson’s disease (PD), and stroke, as well as the potential mechanisms underlying the therapeutic effects of sEH inhibitors.” Moving this sentence from the caption to the main text, would anticipate to the reader the different pathologies covered by the account, and would allow him/her to eventually choose the one of more interest.
  • Parkinson’s disease is characterized by both motor and non-motor symptoms and not only by motor dysfunctions as stated by the authors. Please correct the sentence at line 225. The references cited by the authors highlight this aspect, which in my opinion is even strongly connected with the topic of the review and deserves to be mentioned.
  • Some acronyms some acronyms are not explicit (e.g., poly(I:C), line 178; ANA-12 line 301, SMTP-7 line 312). If the authors believe they are too long to be introduced in the main text, they could add a note.
  • Figure 4 could be a nice Graphical Abstract but is not appropriate for the main text of the account. Conversely, a figure reporting the chemical structures of the sEH inhibitors cited in the text (analogously to Figure 5 of the Conclusion section), such as TPUU, MPTP, AUDA, ANA-12, SMTP-7 and so on should be added for sake of completeness.

Minor:

  • Line 170, the verb is missing: In sum, increased activity of sEH most likely plays a role in the pathogenesis of depression and “suggests” that sEH inhibitors are potential antidepressants.

Author Response

As pointed by the two reviewers, we revised our manuscript carefully. Please find our point-by point responses to the comments raised by the reviewers.
